# Influence of Sweeteners (Sucrose, Sucralose, and Stevia) on Bioactive Compounds in a Model System Study for Citrus–Maqui Beverages

**DOI:** 10.3390/foods11152266

**Published:** 2022-07-29

**Authors:** Francisco J. Salar, Vicente Agulló, Raúl Domínguez-Perles, Cristina García-Viguera

**Affiliations:** 1Phytochemistry and Healthy Food Lab (LabFAS), Department of Food Science and Technology, CEBAS-CSIC, Campus Universitario de Espinardo Edif, 25, 30100 Murcia, Spain; fjsalar@cebas.csic.es (F.J.S.); rdperles@cebas.csic.es (R.D.-P.); cgviguera@cebas.csic.es (C.G.-V.); 2Human Nutrition Unit, Department of Food & Drug, University of Parma, 43125 Parma, Italy

**Keywords:** beverage model system, sweeteners, anthocyanins, vitamin C, flavanones

## Abstract

Recently, new formulations of beverages with low sugar and high bioactive compound contents are being demanded because of their association with metabolic health. However, the sweeteners’ influences on the bioactive compounds remain underexplored. In this sense, this work aims to evaluate the interactions between different bioactive compounds such as flavonoids, vitamin C, and sweeteners (sucrose, stevia, and sucralose) in a functional citrus–maqui beverage. For this purpose, the phytochemical behavior was studied, in model system solutions, during one-month storage at 4 °C. The results obtained corroborated previous descriptions of the interactions between these compounds. However, when studying the bioactives in the model solutions, the loss of flavanones in the citrus solution increased up to 29%, while anthocyanin losses decreased to 27%. The vitamin C losses ranged from 100% (beverage) to 46% (ascorbic acid solution). Regardless, an influence of the sweeteners was observed. Sucrose reduced the anthocyanin and flavanone losses in both model solutions and the beverage, while sucralose increased flavanone loss. Finally, all sweeteners increased vitamin C degradation (up to 75%) when compared to the model solution. In conclusion, sweeteners added to beverages should be taken into account, depending on the bioactive compounds that should be preserved during storage.

## 1. Introduction

In recent years, the consumption of fresh plant-based foods has decreased, while there has been an increase in the consumption of processed foods. These changes in dietary patterns have led to an increase in the prevalence of different metabolic diseases such as obesity, diabetes, hypertension, and cardiovascular diseases, among others [1,2]. In this sense, different studies have shown that the consumption of sugar-sweetened beverages is one of the main factors contributing to the incidence of different metabolic diseases [3,4]. Prompted by epidemiological evidence, food and beverage manufacturers are currently focusing their activity on the development of new formulations, trying to implement potential replacers for added sugars that provide lower natural sugar content. Regarding this trend, industries base their strategy on the addition of other sweeteners, which could constitute an alternative to minimize the consumption of sugar-sweetened beverages and, thus, counteract the postprandial response [5,6].

Regarding this trend, our group designed a new beverage rich in bioactive compounds, sweetened with non-caloric sweeteners, such as stevia (natural) or sucralose (artificial), or a conventional sweetener such as sucrose. This beverage consisted of a mix of citrus fruits and maqui berry, as excellent sources of healthy bioavailable compounds, mainly flavonoids (anthocyanins and flavanones) and vitamin C, all them with demonstrated biological activity in overweight volunteers [7,8], as well as antinociceptive effects [9,10]. Results also indicated that the sweeteners modified the excretions of flavanones and anthocyanins in urine and the bioavailability of polyphenols [11,12].

Despite this, the influence of the diverse added sweeteners was observed when different processing and storage conditions were applied [13,14,15]. The differences observed between sweeteners in terms of bioactive compounds’ stability, bioavailability, and bioactivity were proposed to be due to specific interactions between the different phytochemicals with each other, in addition to the role of the added sweetener [13,16,17]. Nevertheless, as scientific literature on these interactions is scarce, this hypothesis could not be contrasted. Therefore, it is essential to determine possible interactions among these bioactive compounds and different sweeteners, to minimize losses in the functional properties of beverages developed in the future.

Given these antecedents, the objective of the present work was to study possible interactions among phytochemical compounds (anthocyanins, flavanones, and vitamin C) and the added sweeteners (stevia, sucralose, and sucrose), on the previously developed citrus–maqui beverage. For this purpose, model solutions, consisting of maqui or ascorbic acid (vitamin C), citrus juice, and citrus–maqui beverage, with and without sweeteners added, were studied during 27 days at the storage temperature of 4 °C.

## 2. Materials and Methods

### 2.1. Chemicals and Reagents

Cyanidin 3-*O*-glucoside was obtained from TransMIT (Geiben, Germany), L-Ascorbic acid (AA) from Acros Organics (Morris, NJ, USA), and Dehydroascorbic acid (DHAA), hesperidin, narirutin, and eriocitrin from Sigma-Aldrich (St. Louis, MO, USA). Formic acid and methanol, both of them of HPLC grade, were purchased from Panreac (Barcelona, Spain). Ultrapure water from a Milli-Q Advantage A10 ultrapure water purification system (Millipore, Burlington, MA, USA) was used to prepare all solutions.

### 2.2. Fruits and Sweeteners

Citrus beverages were provided by Cítricos de Murcia S.L. (Ceutí, Murcia, Spain) and AMC Grupo Alimentación S.A. (Espinardo, Murcia, Spain). Fresh organic dry maqui powder was purchased from Maqui New Life S.A. (Santiago, Chile). Sucrose, stevia (as Rebaudioside 98%), and sucralose (98%) were provided by AB Azucarera Iberia S.L. (Madrid, Spain), Agriestevia S.L (Molina de Segura, Murcia, Spain), and Sunvision Sweet Co., Ltd. (Xintai City, China), respectively.

### 2.3. Experimental Design

The citrus–maqui beverage was prepared as previously described by Salar et al. [13]. To create the model system, maqui berry powder and ascorbic acid solutions were prepared in the same concentrations as the original beverage (citrus–maqui base drink) (1 g/100 mL and 23 mg/100 mL, respectively), and dissolved in a citric acid buffer adjusted to pH 3, to reach the same pH of the original drink. The selected sweeteners (stevia, sucralose, and sucrose) were added in the same concentrations as in the base drink (4 mg/100 mL stevia and sucralose and 7.5 g/100 mL sucrose) to the different samples, as previously described by Agulló et al. [10]. The energetic value of the beverages developed with stevia, sucralose, and sucrose was 32, 29, and 74 kcal/100 mL, respectively. A control, without added sweetener, was also considered. Samples were labeled according to Table 1.

Ten aliquots of each reaction sample were kept in screw-top transparent plastic vials (volume 10 mL, 56 mm × 18 mm Ø), and stored in the dark at 4 °C, for 27 days. The pH value was measured for each analysis (pH Electrode 5010; Hach Company, Düsseldorf, Germany). All samples designed and experimental conditions tested were prepared in triplicates (*n* = 3) and all analytical determinations were performed in triplicate (*n* = 3). The sampling was carried out on days 0, 3, 6, 9, 15, and 27.

### 2.4. Quantitative and Qualitative Analysis of Phenolic Compounds

The identification and quantification of phenolic compounds were carried out by applying the previously reported method [13], based on Gironés-Vilaplana et al. [18]. Flavanones and anthocyanins were quantified as hesperidin (wavelength of 280 nm) and cyanidin 3-*O*-glucoside (wavelength of 520 nm), respectively, and expressed as mg/100 mL of sample.

### 2.5. Extraction and Analysis of Vitamin C

Vitamin C content was determined by applying the UHPLC-ESI-QqQ-MS/MS-based method and calculated by comparison with standard curves of ascorbic and dehydroascorbic acids [19]. The results were expressed as mg/100 mL of sample.

### 2.6. Statistical Analyses

Results were presented as means ± SD (*n* = 3). The statistical differences between experimental conditions were retrieved resorting to paired *t*-test to compare two conditions, and analyses of variance (ANOVA) and multiple range test of Tukey to identify differences between three or more conditions. All statistical analyses were performed using SPSS 19.0 software (LEAD Technologies, Inc., Chicago, IL, USA). The level of statistical significance was set at *p* < 0.05.

## 3. Results

### 3.1. Flavonoid and Vitamin C Degradation in Control Solutions

The initial total concentrations for flavonoids were 9.28 and 16.46 mg/100 mL, on average, for flavanones and anthocyanins, respectively, showing the same values as the maqui–citrus based drink (Table 2). The initial content of vitamin C (calculated as the sum of AA and DHAA) for all samples was 23.02 mg/100 mL (AA represented over 95% for all samples), on average (Table 2), and results were expressed as a percentage of loss of these total amounts.

The bioactive compound degradations, in the citrus–maqui control during storage (Figure 1), were in concordance with those previously obtained for the citrus–maqui beverage [13,14], resulting in a 10% loss of total flavanones (final concentration of 8.36 mg/100 mL; characteristic of citrus juices), 35% loss of total anthocyanins (final concentration of 10.66 mg/100 mL; characteristic of maqui berry), and 100% loss of vitamin C (also from citrus juice although for this the total loss was observed at day 27 relative (18 days later than the citrus–maqui control)).

On the other hand, regarding these losses, some differences were observed when the compounds were studied separately in model solutions. Regarding the stability of flavanones, for citrus control, the final loss of flavanones during the study increased up to 29% (final concentration of 6.59 mg/100 mL), while in the maqui control solution the total loss of anthocyanins presented a slight decrease to 27% (final concentration of 12.02 mg/100 mL) at day 27 (Figure 1A,B, respectively). Hence, the lower degradation of the flavanones observed in the citrus–maqui beverage could be due to a possible protective effect resulting from the interaction of the anthocyanins with flavanones, by inter-molecular co-pigmentation, according to previous reports [20].

On the other hand, the higher loss of anthocyanins in the control beverage (up to 35%), compared to maqui solution (up to 27%), could be explained by the fact that these colored flavonoids are easily degraded by other compounds present in the beverage matrix, such as ascorbic acid, different enzymes, etc. [21,22,23]. This explanation is reinforced because, in the model system, other environmental factors (pH, temperature, light, etc.) are minimized. Additionally, it might be considered that anthocyanin degradation could be partially reduced as a consequence of the presence of flavanones in the citrus–maqui control, conversely to the maqui control beverage, which could be related to the improvement of their stability as previously discussed [20]. Due to this, it is important to study the interaction with ascorbic acid, as will be discussed below.

Beyond the content of flavanones and anthocyanins, the citrus–maqui beverage is also a valuable source of vitamin C [13]. The vitamin C concentration during storage, as a result of the interaction with other beverage components, needs to be addressed as well. In this regard, it has been widely reported that ascorbic acid is affected by several intrinsic and extrinsic factors in juices during storage, such as temperature, oxygen, light exposure, metal ions, or pH [24,25]. In this frame, citrus–maqui control beverages presented a faster degradation (100% loss at day 9) in comparison with citrus controls (100% loss at day 27) or ascorbic acid model solutions (46% total loss during 27 days of storage; final concentration of 12.43 mg/100 mL) (Figure 1C). Because of the lower degradation in the vitamin C control relative to the citrus and citrus–maqui controls, it could be speculated that the degradation of ascorbic and dehydroascorbic acids is due, regardless of extrinsic factors, to endogenous enzymes with oxidase activity in the matrix of the citrus juice [26], such as polyphenol oxidase, peroxidase, and especially ascorbate oxidase, which catalyze ascorbic acid oxidation [14,27,28]. Last, but not least, the faster degradation of vitamin C, present in citrus–maqui beverages seems to be not only due to extrinsic factors, or to the presence of endogenous enzymes, but also to the presence of the anthocyanins characteristic of the maqui fruit. This point has been previously studied, concluding that interactions of ascorbic acid with anthocyanins led to mutual degradation, due to different mechanisms, such as direct condensation of ascorbic acid on C_4_ of the anthocyanin molecule, or the oxidative cleavage of the pyrilium ring by free radicals generated by ascorbic acid or the auto-oxidation of ascorbic acid, as a consequence of the release of free radicals mechanism [29,30,31,32].

### 3.2. Interaction between Bioactive Compounds and Sweeteners

As previously mentioned, the stability of the bioactive compounds in the beverages might be also affected by the presence of sugars in the beverage matrix. Therefore, it is important to determine possible interactions of the different sweeteners used by the beverage industry, with the functional phytochemicals present in these drinks.

#### 3.2.1. Interaction between Flavonoids and Sweeteners

Results demonstrated that sweeteners have an influence on the individual bioactive compounds of the citrus–maqui beverages (Figure 2A). Thus, flavanone total losses at the end of the studied period ranged from 5% loss (final concentration of 8.82 mg/100 mL) when sucrose was added to 20% (final concentration of 7.42 mg/100 mL) when sucralose was added. Intermediate loss values were recorded when no sweetener or stevia was added (10% loss; final concentration of 8.36 and 8.51 mg/100 mL, for control and stevia sweetened beverage, respectively).

Regarding anthocyanins (Figure 2B), all sweeteners showed a different protective effect against the degradation of these bioactive compounds, when compared to the unsweetened control (35% loss; final concentration of 10.70 mg/100 mL). Thus, the anthocyanin losses decreased with the different sweeteners as follows: sucralose (32% loss; final concentration of 11.19 mg/100 mL) > sucrose (25% loss; final concentration of 12.35 mg/100 mL) > stevia (19% loss; final concentration of 12.31 mg/100 mL).

To understand the influence of the separate sweeteners used on individual bioactive flavonoids (flavanones and anthocyanins), all sweeteners were compared one to another and with the control without sweetener (Figure 3 and Figure 4, respectively). In this sense, the losses of flavanones, in the citrus samples, varied from 5 to 30% (final concentration of 8.82–6.50 mg/100 mL), at the end of the study, depending on the added sweetener. It should be noted that, again, those with added sucralose presented similar losses relative to those observed in citrus control made without sweetener (approx. 30% loss; final concentration of 6.50 mg/100 mL). Nonetheless, sucrose rendered a protective effect, decreasing the percentage of losses up to 5% (final concentration of 8.82 mg/100 mL). Those citrus beverages sweetened with stevia presented intermediate values (19% loss; final concentration of 7.52 mg/100 mL).

Furthermore, it should be noted that, in the citrus samples, all sweeteners showed relatively similar behavior related to citrus–maqui beverages (Figure 2), where those with sucralose added exhibited similar losses as when no sweeteners were added. On the other hand, when adding sucrose the lowest flavanones loss was obtained (5%; final concentration of 8.82 mg/100 mL).

The scientific literature on the mechanism of flavanone degradation during storage using sweeteners is scarce; however, these results confirm previous ones [13], where a similar behavior was found in terms of flavanone loss for samples stored under refrigerated conditions. In this regard, sucrose seems to be the best sweetener for protecting flavanones in citrus juices, while sucralose would not be recommended. These results would be explained by taking into consideration the inhibitory effect on oxidative enzymes (polyphenol oxidase and peroxidase), previously demonstrated by stevia or sucrose. This path would contribute to prevent the polyphenols’ degradation dependent on these enzymes [33,34].

Similarly, when the interaction of anthocyanins, in maqui control solutions, with sweeteners, was determined, a preservative effect of the sweeteners was observed, with final losses of 16%, 22%, and 25%, representing a final concentration of 13.82, 12.84, and 12.35 mg/100 mL, for sucrose, sucralose, and stevia, respectively (Figure 4), compared to maqui control (27% loss; final concentration of 12.02 mg/100 mL). In general, the losses were lower in the maqui beverages, where anthocyanins are present (5–20%) than in the citrus juice (5–30%). In this sense, similar results were observed by Gironés et al. [18], who attributed this result to the stabilization process of maqui anthocyanins on the hesperidin of citrus fruits. Additionally, as observed with flavanones, sucrose was the sweetener that provided the best results, while sucralose and stevia did not differ much from the unsweetened control, as previously reported by Wozniak et al. [35]. This protective effect of sucrose on anthocyanins has been previously reported by other authors [36,37].

Nevertheless, the percentage of loss of anthocyanins in maqui solutions was lower than in the citrus–maqui beverages (16–27% vs. 20–35%, respectively), at the end of the study. This higher degradation of anthocyanins, regardless of added sweetener, could be attributed to the presence of vitamin C, as described above.

#### 3.2.2. Interaction between Vitamin C and Sweeteners

Due to its functional interest, as well as its interactions with other phenolic compounds, vitamin C was also considered for this study, to obtain a complete picture of the influence of the diverse sweeteners on this bioactive nutrient, in the designed citrus–maqui beverage. In this sense, the effect of sweeteners on vitamin C in different beverage matrices has also been previously studied by other authors [35,38].

It must be mentioned that this is the most unstable bioactive compound of the beverage, as, in general, it presents a 100% loss after less than one month of storage, which is in good agreement with previous studies [14,39,40]. Therefore, in the present work, it was observed that, on day 27, the degradation of vitamin C was complete in the citrus–maqui control and citrus juice (Figure 5A,B), but not when ascorbic acid was studied as a model solution (reaching only 40% loss), as described in Figure 5C.

From these results, the contribution of the sweeteners to the stability of vitamin C was assessed. In this regard, the addition of sweeteners slowed around one week the speed of loss, during the storage, in the citrus–maqui base beverage (Figure 5). Nevertheless, not all samples behaved in the same way. In this sense, sucrose was the least effective in maintaining the concentration of vitamin C in citrus–maqui beverages, as up to 80% total loss (final concentration of 4.60 mg/100 mL) occurred on day 9 when stevia and sucralose still preserved almost 50% (final concentration of 10.82 and 11.31 mg/100 mL) of vitamin C (Figure 5A). These results could be explained by taking into account that sucrose was one of the sweeteners with the highest protective effect on anthocyanins, enabling a major interaction between these compounds, triggering a higher percentage of the breakdown of vitamin C. These results are in full agreement with those previously obtained by Salar et al. [13,14] for this citrus–maqui beverage, processed under industrial conditions.

This interaction with anthocyanins was corroborated when the loss of this vitamin was measured, in the citrus juice sample (Figure 5B). In this case, even if a total degradation was reached after the storage period, the kinetics were slowed down (less than 40% loss after 9 days of storage; final concentration of 17.03 mg/100 mL). It is mentionable that sucralose (72% loss at day 15; final concentration of 6.45 mg/100 mL) and stevia (87% at day 15; final concentration of 2.99 mg/100 mL) increased the degradation rate of vitamin C in citrus juice, while the samples sweetened with sucrose presented the same behavior as the control citrus juice (approx. 50% loss at day 15; final concentration of 10.85 mg/100 mL). These results complement those obtained by other authors, who compared the ability of sucrose, glucose, and lactose in maintaining the vitamin C [41], as we provide results about the effect of two additional sweeteners on vitamin C preservation.

Finally, when studying the influence of the addition of sweeteners on vitamin C in the model solution (Figure 5C), without the interference of other factors present in the beverages, lower losses were observed in all cases. As previously mentioned, vitamin C degradation reached less than 50% (final concentration of 12.43 mg/100 mL) at the end of the study, in the model solution, while greater losses were observed when sweeteners were added (75% on average; final concentration of 5.76 mg/100 mL). Therefore, it could be concluded that there is a degradative activity of sweeteners on vitamin C, independent of the sweetener used. Nevertheless, samples containing stevia showed a slight increase in the degradation speed after day 9, when compared to sucrose or sucralose. Conversely, other authors have reported a protective effect of stevia on vitamin C, with a decrease in the rate of degradation in this vitamin over time [38,42].

These results indicate that the degradation of vitamin C is accelerated due to the presence of any other compound in a food matrix, something that the beverage industry should take into consideration when formulating a new juice or drink.

## 4. Conclusions

In this study, phytochemical compound (naturally present in fruit drinks) interactions were confirmed. In this sense, anthocyanins and vitamin C are key responsible for their mutual degradation, together with the action of endogenous enzymes in citrus juices, that also cause vitamin C degradation. Conversely, a mutual protective effect was demonstrated between flavanones and anthocyanins.

Despite this, it was observed that sucrose presents a protective effect against the degradation of flavonoids, such as flavanones and anthocyanins. Furthermore, due to similar flavanone degradation presented when sucralose is added, when compared to citrus control samples, this sweetener should not be recommended, for citrus beverages, while stevia could be a non-caloric alternative, even if less effective than sucrose. This might be extrapolated for those drinks derived from fruits rich in anthocyanins (such as berries).

Nevertheless, a consensus should be reached if the focus is to extend the shelf life of ascorbic acid, in beverages rich in anthocyanins, as sucrose would not be fully recommended, even if this vitamin shelf life does not extend beyond 10 days even under refrigerated conditions.

In general, sucralose should be avoided, and stevia could be considered as a potential alternative natural non-caloric sweetener, for the beverage processing industry, if developing a new drink rich in flavonoids, reducing somehow the consumption of sugar and taking advantage of the biological attributions of polyphenols and vitamin C.

Finally, it is important to highlight that, depending on the type of matrix and which bioactive compounds have to be preserved, an exhaustive review of the sweeteners should be carried out before drink production.

## Figures and Tables

**Figure 1 foods-11-02266-f001:**
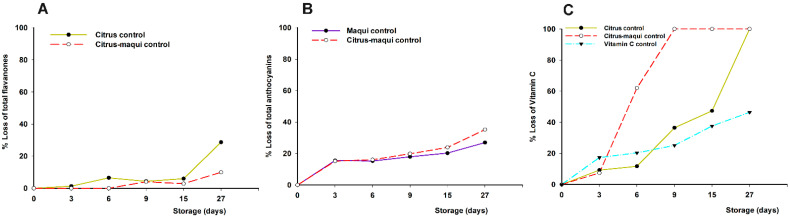
Percentage of loss of bioactive compounds (flavanones (**A**), anthocyanins (**B**), and vitamin C (**C**)) in citrus–maqui control beverages, citrus juice (citrus control), maqui control solution, and ascorbic acid control solution, measured during 27 days storage at 4 °C. Significantly different final concentration absolute values according to statistical analysis (*t*-test or ANOVA and Tukey’s multiple rank test) were found *p* < 0.01 and *p* < 0.001, for (**A**,**C**), respectively, while no significant differences were found for (**B**).

**Figure 2 foods-11-02266-f002:**
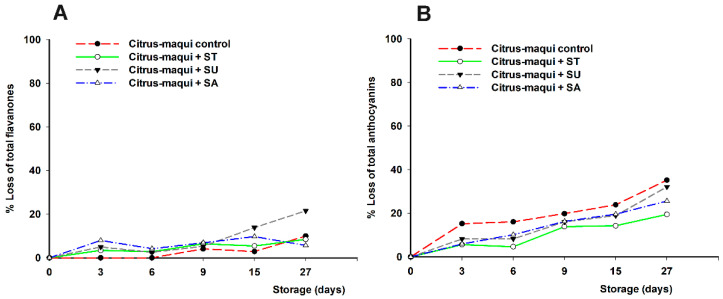
Percentage of loss of total flavanones (**A**) and anthocyanins (**B**) in citrus–maqui beverage samples (control without sweeteners), and citrus–maqui beverage samples plus sweeteners, stevia (ST), sucralose (SU), or sucrose (SA), measured during 27 days at 4 °C. Significantly different final concentration absolute values according to statistical analysis (ANOVA and Tukey’s multiple rank test) were found at *p* < 0.05, for both figures.

**Figure 3 foods-11-02266-f003:**
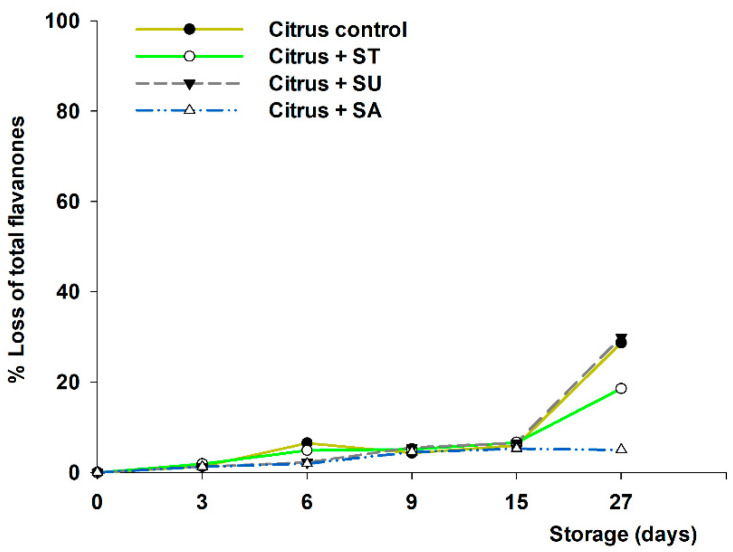
Percentage of loss of total flavanones in citrus fruit beverage samples (control without sweeteners), and citrus fruit beverage samples plus sweeteners, stevia (ST), sucralose (SU), and sucrose (SA), measured during 27 days at 4 °C. Significantly different final concentrations absolute values according to statistical analysis (ANOVA and Tukey’s multiple rank test) were found at *p* < 0.01.

**Figure 4 foods-11-02266-f004:**
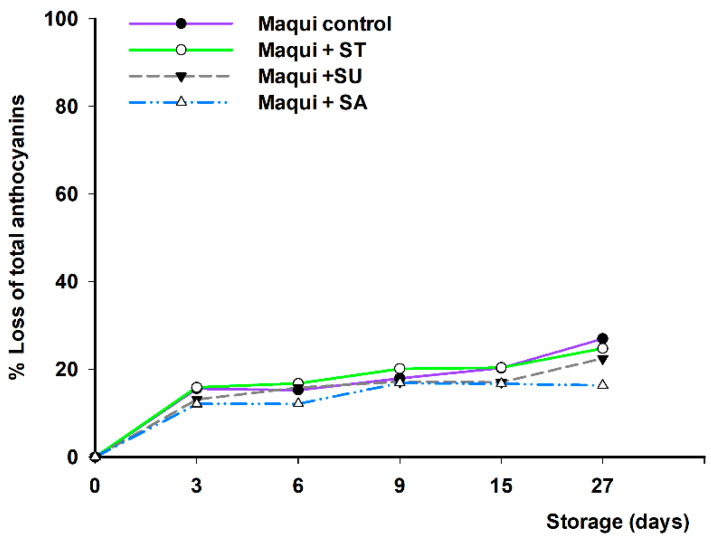
Percentage of loss of total anthocyanins in maqui-berry beverage samples (control without sweeteners) and maqui-berry beverage samples plus sweeteners, stevia (ST), sucralose (SU), and sucrose (SA), measured during storage for 27 days at 4 °C. Significantly different final concentration absolute values according to statistical analysis (ANOVA and Tukey’s multiple rank test) were found at *p* < 0.01.

**Figure 5 foods-11-02266-f005:**
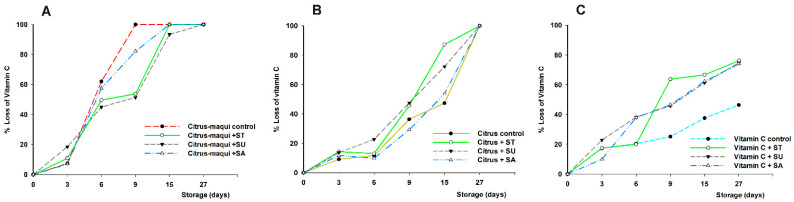
Percentage of loss of vitamin C in citrus–maqui beverage (**A**), citrus juice (**B**), and vitamin C solution (**C**), with added sweeteners stevia (ST), sucralose (SU), and sucrose (SA) or without (control), measured during storage for 27 days at 4 °C. Significantly different concentrations’ absolute values at days 9, 15, and 27, for (**A**–**C**), respectively, according to statistical analysis (ANOVA and Tukey’s multiple rank test) were found at *p* < 0.001.

**Table 1 foods-11-02266-t001:** Codification of samples included in the experimental design.

Code	Sample
Citrus control	Citrus juice without sweeteners added
Citrus control + ST	Citrus juice sweetened with stevia
Citrus control + SU	Citrus juice sweetened with sucralose
Citrus control + SA	Citrus juice sweetened with sucrose
Maqui control	Maqui solution without sweeteners added
Maqui control + ST	Maqui solution sweetened with stevia
Maqui control + SU	Maqui solution sweetened with sucralose
Maqui control + SA	Maqui solution sweetened with sucrose
Vitamin C control	Ascorbic acid solution without sweeteners added
Vitamin C control + ST	Ascorbic acid solution sweetened with stevia
Vitamin C control + SU	Ascorbic acid solution sweetened with sucralose
Vitamin C control + SA	Ascorbic acid solution sweetened with sucrose
Citrus–maqui control	Citrus–maqui sample (base drink) without sweeteners added
Citrus–maqui control + ST	Citrus–maqui sample (base drink) sweetened with stevia
Citrus–maqui control + SU	Citrus–maqui sample (base drink) sweetened with sucralose
Citrus–maqui control + SA	Citrus–maqui sample (base drink) sweetened with sucrose

**Table 2 foods-11-02266-t002:** Initial concentration (mg/100 mL) of bioactive compounds in the control solutions and maqui–citrus base drink.

**Flavanones**	**Eriodictyol 7-*O-*Rutinoside (Eriocitrin*)***		**Naringenin 7-*O-*Rutinoside** **(Narirutin)**		**Hesperetin 7-*O-*Rutinoside (Hesperidin)**		**Total**
	2.81 ± 0.05		2.44 ± 0.02		4.03 ± 0.06		9.28 ± 0.09
**Anthocyanins ^z^**	**Dp 3*-O-*sam-5*-O-*glc**	**Dp 3,5*-O-*diglc**	**Cy 3,5*-O-*diglc** **+** **Cy 3*-O-*sam-5*-O-*glc**	**Dp 3*-O-*sam**	**Dp 3*-O-*glc**	**Cy 3*-O-*sam** **+** **Cy 3*-O-*glc**	**Total**
	4.76 ± 0.02	4.66 ± 1.13	1.96 ± 0.01	1.24 ± 0.00	2.81 ± 0.01	1.04 ± 0.00	16.46 ± 1.11
**Vitamin C**	**Ascorbic acid**			**Dehydroascorbic acid**			**Total**
	21.39 ± 0.37			1.63 ± 0.05			23.02 ± 0.42

^Z^ Cy, cyanidin; Dp, delphinidin; Glc, glucoside; Sam, sambubioside. Bold refers to the name of the different bioactive compounds.

## Data Availability

The data are available from the corresponding author.

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
