# Peer review of "Influence of Sweeteners (Sucrose, Sucralose, and Stevia) on Bioactive Compounds in a Model System Study for Citrus–Maqui Beverages"

_foods, 2022, doi:10.3390/foods11152266_

Round 1

Reviewer 1 Report

ARTICLE REVIEW

Title: Influence of sweeteners on bioactive compounds in a model system study

Authors: Francisco J. Salar, Vicente Agulló, Raúl Domínguez-Perles and Cristina García-Viguera

 New formulations of beverages with low sugar and high bioactive compound contents are being demanded because of their association with metabolic health. The sweeteners' influence on the bioactive compounds remains underexplored. In this sense, this work aims to evaluate the interactions between different bioactive compounds such as flavonoids, vitamin C and sweeteners (sucrose, stevia, and sucralose) in a functional citrus-maqui beverage. For this purpose, the phytochemicals behaviour was studied, in model system solutions, during one month storage, at 4 °C. The results obtained corroborated previous descriptions of the interactions between these compounds. However, when studying the bioactives in the model solutions, the loss of flavanones in the citrus solution increased up to 29%, while anthocyanin losses decreased to 27%. The vitamin C losses ranged from 100% (beverage) to 46% (ascorbic acid solution). Despite, an influence of the sweeteners was observed. Sucrose reduced the anthocyanins and flavanones losses in both model solutions and the beverage, while sucralose increased flavanones loss. Finally, all sweeteners increased vitamin C degradation (up to 75%) when compared to the model solution.

The worki is well written. The text clear and easy to read.

The conclusions are consistent with the evidence and arguments presented. The conclusions respond to the set goal of the work.

Comments:

Line 40: it is difficult to talk about a healthy drink in the context of the addition of sucrose

The analytical methods used in the research are described very laconically.

Author Response

Thank you very much for your considerations and suggestions that allow us to improve the manuscript. 

Regarding the point-by-point response:

Line 40: it is difficult to talk about a healthy drink in the context of the addition of sucrose.

Following the reviewer suggestion, this statement was re-drafted in the reviewed version of the MS (page 1 of the reviewed version of the MS).

The analytical methods used in the research are described very laconically.

The analytical methods were described briefly to overcome plagiarism issues. However, to avoid the presentation of insufficient information according to the standards of scientific publications dedicated bibliographic references containing all information needed to reproduce the methodology used have been provided throughout the whole manuscript.

Reviewer 2 Report

In the manuscript submitted for comment, the authors studied new beverage formulations that are low in sugar and high in bioactive compounds. The study is not entirely new. A review of the literature , e.g. Moldovan, B. and David L. 2020, investigates the degradation process of anthocyanins in Cornelia cherry juice supplemented with different sweeteners (sucrose, fructose) or artificial sweeteners (aspartame and acesulfame potassium) for 6 days. Therefore, in my opinion, the overall presentation can be improved, especially with the determination of the kinetic parameters - rate constants (K) and half-time values ( t1/2) of anthocyanins and vitamin C degradation processes that may affect the shelf life of citrus and maqui drinks.

In Table 1, the coding of the samples included in the experimental design should be in text form.

The quantification of the bioactive compounds, the authors, use spectrophotometric methods to quantify flavanones as hesperidin at 280 nm, and anthocyanins as cyanidin 3-O-glucoside at 520 nm. However, they quantify vitamin C concentration using UHPLC-ESI-QQQ MS/MS. Thus, explain how they can quantify, in citrus extract, by spectrometry only the hesperidin? And in the maqui extract by the same method, only 3-O-glucoside? Are there no other compounds in the extracts that might contribute to these absorptions? Why didn't the authors use the second method to identify and quantify all the compounds?

In Table 2 the authors present the initial concentration (mg/100 mL) of bioactive compounds in the maqui-citrus base drink, but how were the flavanones (eriocitrin, narirutin, and hesperidin) identified and quantified? since the authors did not use UHPLC-ESI-QQQ MS/MS to identify these compounds. Also, why did they not also present the final concentrations of the compounds, after the stability test?

Author Response

Thank you very much for your considerations and suggestions that allow us to improve the manuscript. 

Regarding the point-by-point response, please see the attachment.

Reviewer 3 Report

Comments to the Authors

Manuscript ID: foods-1787238

General remarks:

The authors studied the influence of three sweeteners (natural, conventional and artificial) on degradation of flavanones, anthocyanins and vitamin C in a citrus based beverage. Performed analysis are relevant in the field however great addition to the manuscript would be the inclusion of concrete energetic value for each Citrus -maqui beverage sample and used sweetener in terms of their health promoting effect. This is my main objection, which is of most relevance regarding the statements presented in the introduction and overall objective of the work.

Figures resolution needs to be improved for clearer visual appearance as well as the scaling on y-axis.

The specific comments are listed below.

Title

Please include content in the title. Furthermore, the authors should include the name of the beverage since the analysis were performed also in the beverage itself not only in model systems. It would be beneficial to more closely define the sweeteners used also, such as natural, artificial and conventional or just state the used sweeteners so that readers can see from the title content of the manuscript.

Abstract

The abstract is well written and reflects all performed analysis and main results.

Line 16: comma in front of the storage temperature is unnecessary please remove.

Manuscript

Lines 41: The sentence needs rephrasing for clarity. Better would be and conventional sweetener such as sucrose. Please improve.

Line 46: Please use only two most relevant references.

Line 54: Please improve to of beverages developed in the future. 

Line 50: Please include the reference corresponding to the previously developed beverage.

Line 61: Please improve the sentence. Suggestion during 27 days under storage temperature 4°C.

Line 80: Please state the pH and concentration of the original beverage.  

Line 81: Please include separately dissolved.

Line 90: Please include the electrode used for pH measurement.

Line 95: Better would be previously reported method. Please improve.

Line 97: Instead of per in the concentration units please use /  as previously in the experimental design section. Please carefully check the entire manuscript and revise this (lines 110, 112…).

Line 105: the word applied seems like excessive. Please exclude. Furthermore, I supposed that under condition authors meant comparison of samples. Please improve for clarity.

Line 116: Please clarify are the results presented within table 2 average results of all control samples (Citrus control, Maqui control, Vitamin C control, and Citrus-maqui control) as stated in the text above or only for Citrus-maqui control as included in the table 2 title and revise accordingly.  

Line 119: Previously studied is unnecessary. Please correct.

Line 120: According to the graph 1B it seems like melioration of the anthocyanins content occurred in Maqui control i.e. the loss is reduced in this sample. That was also stated in the abstract and in the following text (line 131). Please check and clarify.

Line 130: Please state that this corresponds to day 27.

Line 131: Please exclude d from the decreased.

Line 142: Instead of which are absent in better would be conversely to the maqui control beverage Please revise.

Line 151: Please check according to the graph for citrus control 100% loss of vitamin C occurred nearly to day 27.

Line 164: Please use C4 instead of carbon 4

Line 172: Better would be have an influence on the

Line 187: This sentence needs revision in order to provide clarity. I suppose that authors wanted to compare control model samples with those with sweetener included.

Line 197: Please state the sweeteners according to order of appearance in the graph legend.

Line 201: Here also, please state the sweeteners according to order of appearance in the graph legend.

Line 204: I will suggest to include relatively similar behaviour  because there are differences for flavanones.  Please put in uppercase the adequate parts in the units (cm-1).

Line 206: Please revise to the lowest flavanone loss was obtained.

Line 210: please use another word instead of veine for example in this regard.

Line 214: Please use prevent instead of preserve since it better fits to the sentence context.

Line 234: The authors stated that the beverage has formulated according to the previous study so it is not new.

Line 238: Please add which is in good agreement…

Line 241: Please include 5A and 5B instead of only figure 5.

Line 242: Better would be as visible in Figure 5C but also in Figure 1C.

Line 246: Here also, please state the sweeteners according to order of appearance in the graph legend.

Line 265: Here in citrus model sample, sucrose exhibited the most protective effect on vitamin C not the least. That is contradictory to statement in the rest of the sentence, please revise. Maybe in previous study sucrose exhibited the least protective effect compared to glucose and lactose, but that is not in line with the effect provided in this study.

Conclusion

Well written, summarizing the main outcomes of the study and with appropriate recommendations based on the outcomes.

Figures

Pictures in better resolution are to be provided by the authors for all figures.

Figure 1. On the y-axis is stated loss of flavanones or anthocyanins while on the rest of the figures is stated total loss. Please check.

Figure 2, 3 and 4. Please use different scaling on the y-axis (for example up to 60%) for better visibility and recognition of different samples.

Author Response

(The authors gave the same response as above.)

Round 2

Reviewer 2 Report

I agree with the justifications and changes made by the authors, however, in my opinion, the experimental design should have been carried out more carefully and the references to previous studies presented more clearly.

Author Response

Thank you very much for your considerations and suggestions that allow us to improve the manuscript.

We do agree that the design could be done in a different way, in order to look for different objectives, as degradation kinetics, etc. Nevertheless, the main objective was to understand the interaction between the different phytochemicals present in the beverage, not their individual behavior, as has been accepted by the other referees. However, we are working on other experiments related to kinetic formulas, comparing different processing technologies, but, as we have said, this was not the main objective of the present work.

In addition, the previous studies described in the experimental design has been introduced clearier in the manuscript following the reviewer suggestions.

Reviewer 3 Report

The authors performed most of the proposed sugesstions which contributed to the better understanding of the manuscript.

Author Response

Dear reviewer, we wish to thank for all your comments, that have highly improve the present article.